# Magnitude of ectopic pregnancy, management methods, and its associated factors among pregnant women attending Ambo University Referral Hospital in Oromia Regional State, Ethiopia: A seven years retrospective institutional based cross-sectional study

Nigussie Tesfaye Gizaw[1]*, Mekbeb Afework K/Mariam[2], Motuma Gutu Fayera[3]

1 Ambo University College of Health Sciences and Referral Hospital Department of Anatomy, Ambo, Ethiopia, 2 Addis Ababa University College of Health Sciences Department of Anatomy, Addis Ababa, Ethiopia, 3 Ambo University College of Health Sciences and Referral Hospital Department of Gynecology and Obstetrics, Ambo, Ethiopia

* nigetesfaye14@gmail.com

## Abstract

### Background

Ectopic pregnancy is the implantation of a blastocyst outside of the endometrial lining of the uterus. Ectopic pregnancy can take several forms, including ovarian, abdominal, and tubal. The most prevalent place for ectopic pregnancy is the fallopian tube, which accounts for over 97.7% of all ectopic gestations. The ampulla accounts for around 80% of tubal pregnancy, followed by the isthmus (12%), fimbria (5%). Ectopic pregnancy is diagnosed with the classic triad of amenorrhea, abdominal pain, vaginal bleeding, and a positive pregnancy test.

### Objective

This study aimed to assess the magnitude of ectopic pregnancy, its management outcome, and associated factors among pregnant women attending Ambo University Referral Hospital in the Oromia Region, Ethiopia, in 2024.

### Methods

A seven-year retrospective cross-sectional study, from February 2018 to April 2024, was conducted at the Ambo University Referral Hospital, which is located in Ambo town, Ethiopia. Data concerning all pregnant mothers who were admitted and managed for ectopic pregnancy were extracted from the medical records of patients and the operation book by a trained data collector through Kobo Toolbox electronic data collection software. The collected data was checked first for its completeness, and it

**Data availability statement:** All relevant data are within the manuscript and its Supporting information files.

**Funding:** The authors received no specific funding for this work.

**Competing interests:** The authors have declared that no competing interests exist

was exported into SPSS version 26 software for data analysis. Then Descriptive statistics were employed for summarizing the data, and bivariate and multivariate logistic regression analyses were used to identify the independent effect of the predictor variable on the outcome variable.

## Results

From February 2018 to April 2024, there were 17,687 total pregnancies, 6,249 gynecologic admissions, and 182 cases of ectopic pregnancies at Ambo University Referral Hospital. A total of 173 patients with ectopic pregnancy were included in the data analysis. The magnitude of ectopic pregnancy was 0.98% among the total pregnancies and accounted for 2.77% of all gynecological admissions during the study period. Most of the patients, 81 (46.8%), were in the 25–29 years age group, with a mean age of 27.16±SD 4.77 years. Mothers who had a previous history of abortion, a history of pelvic inflammatory disease, a history of ectopic pregnancy, or a history of tubal surgery had a statistically significant association with ectopic pregnancy. The majority of the patients were married, 98 (56.6%), and urban residents, 121 (69.9%). The majority of ectopic pregnancies occurred on the right side of the fallopian tube 144 (83.24%). Among the majority of ectopic pregnancies, 159 (61.3%) were tubal ampullary ectopic pregnancies.

## Conclusion and recommendation

The major risk factors identified in this study were previous abortion, pelvic inflammatory disease, a previous history of ectopic pregnancy, and previous tubal surgery. The magnitude of the ectopic pregnancy in this study was 0.98%, which is similar to the global range. The majority of ectopic pregnancies occurred on the right side of the fallopian tube 144 (83.24%) and 160 (92.49%) were ruptured. Further research is needed to assess why ectopic pregnancy is most common in the right fallopian tube.

## Introduction

Ectopic pregnancy (EP) is a gynecological emergency in which a blastocyst implanted outside the endometrial lining of the uterus. There are two types of ectopic pregnancies: tubal and nontubal. Nontubal pregnancies occur when oocytes are fertilized outside of the fallopian tube or when they are fertilized inside the tube but protrude into the peritoneum. These abnormal sites of implantation include the ovaries, the abdominal cavity or the mesentery, and the uterine tube [1]. Nearly 97.7% of all ectopic gestations occur in the fallopian tube, which is the most common location for ectopic pregnancy. Approximately 80% of tubal pregnancies occur in the ampulla, with the isthmus (12%), and fimbria (5%), [1,2]. Ectopic pregnancy is diagnosed with the classic triad of amenorrhea, abdominal pain, vaginal bleeding, and a positive pregnancy test. However, only 50% of patients exhibit the usual symptoms at presentation [3]. The specific cause of ectopic pregnancy remains

unknown. However, women are susceptible to EP, for several reasons. major risk factors and causes of pelvic infection, including the use of contemporary contraceptives, appendicitis, puerperal sepsis, abortion, and pelvic inflammatory disease (PID) [4,5]. Ectopic pregnancy is more likely if the uterine tubes are malformed or damaged from prior surgery or infection, tumors, or, in rare cases, birth defects. There is a greater chance of infertility and tubal ectopic pregnancy (EP) among women who have experienced tubal ectopic pregnancy (EP).. Abdominal or transvaginal ultrasound is an important tool in the diagnosis of ectopic pregnancy [6,7]. According to a retrospective study carried out in India, younger individuals are disproportionately affected by ectopic pregnancy. The incidence of ectopic pregnancy was higher in multiparous women (61.1%) than in primigravidae (19.4%) [8]. According to a five-year retrospective study conducted at Benin, the most affected age group was between 20 and 29 years [9]. A case-control study conducted in India on 220 cases revealed that 8% had a history of smoking and, study conducted in China concluded that the smoking can cause EP by causing transport malfunction and an aberrant microenvironment in the fallopian tubes [10,11]. However, a retrospective study in India revealed no significant association with smoking [12].According to study conducted in Egypt, the prevalence of ectopic pregnancy was found to be 0.52%, 0.62%, and 0.72% in 2018, 2019, and 2020, respectively, along with the related risk factors [13]. A study conducted on the determinants of ectopic pregnancy in the southwestern Ethiopia reported that being single was an independent predictor of ectopic pregnancy [14]. EP is the principal cause of maternal mortality and morbidity worldwide [15]. Worldwide, approximately 1_2% of all naturally conceived pregnancies result in ectopic pregnancy [14][16]. Studies in Africa also revealed a similar incidence of 1–2% of ectopic pregnancies [17]. In Ethiopia, the magnitude of ectopic pregnancy among total deliveries is 0.82% [4]. Ectopic pregnancy have significant complications. The common complications of ectopic pregnancy are massive hemorrhage, shock, and adverse psycho-emotional effects on the mother, and also affects the subsequent fertility of women [18]. The purpose of this study was to determine the risk factors, magnitude of ectopic pregnancy, and management outcome of ectopic pregnancy among pregnant patients who visit Ambo University referral hospitals in Oromia Regional State, Ethiopia.

## Conceptual framework

This conceptual framework was constructed after multiple relevant literature review were reviewed to show the factors that affect the ectopic pregnancies (Fig 1).

## Objectives

### General objectives

- The main aim of this study was to assess the magnitude of ectopic pregnancy, its management outcome, and associated factors among pregnant women attending Ambo University Referral Hospital in Oromia Regional State, Ethiopia, from February 2018 to April 2024.

### Specific objectives

- To determine the magnitude of ectopic pregnancy among pregnant women attending Ambo University Referral Hospital in Oromia Regional State, Ethiopia, in 2024.

- To assess the management outcome of ectopic pregnancy among pregnant women attending Ambo University Referral Hospital in Oromia Regional State, Ethiopia, in 2024.

- To identify the factors associated with ectopic pregnancy among pregnant women attending Ambo University Referral Hospital in Oromia Regional State, Ethiopia, in 2024

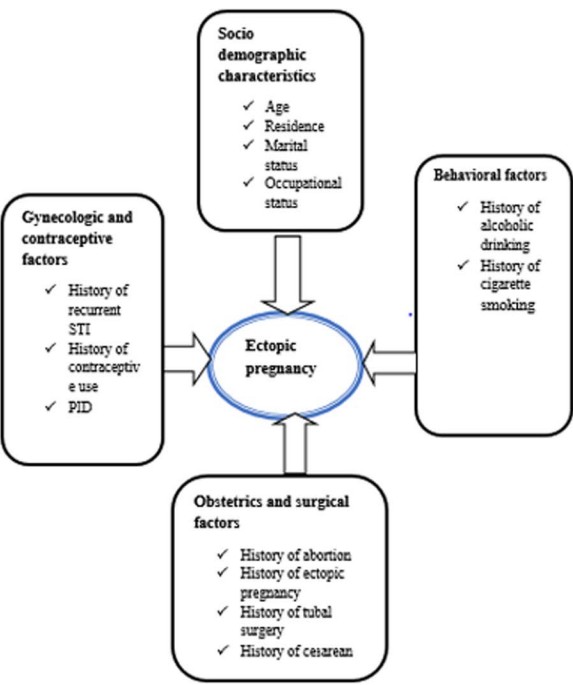

**Fig 1. This conceptual framework was formulated after several relevant literature reviews were reviewed.**

## Methods and materials

### Study areas, design and period

The study was conducted in the AURH West Shewa Zone, Oromia Regional State, Ethiopia. AURH was purposefully selected because it is the only large teaching and referral hospital in the west Shewa zone and provides service for the entire zone and some parts of nearby zones. It has a high case flow and receives many referrals from different hospitals located within the zones. AURH is located within Ambo city, which has a latitude and longitude of 8º59'N, 37º51'E, and an elevation of 2,101 m, which is 126 km west of Addis Ababa, Ethiopia. It started as a referral hospital in 2016. AURH has 14 gynecologists and 40 midwives; the gynecologic ward has 24 beds; and the labor ward has 14 beds (AURH human resources directorate). This hospital provides general and specialized clinical services, including ANC, family planning, delivery services, and treatment or obstetric complications, which are some of the services provided in gynecologic and obstetric wards. All total pregnancies and gynecological surgeries carried out within the past seven years (from February 2018 to April 2024) were 17,687 and 6,249, respectively (AURH HMIS).

### Study design, period and population

An institutional-based retrospective cross-sectional study was conducted from February 1 to April 21, 2024.The source of the population was all pregnant women admitted to the MCH, Gynecology and Obstetrics Department of AURH in the past seven years (from February 2018 to April 2024).The study population was all pregnant women who had been admitted to the MCH, Gynecology and Obstetrics Department of AURH with a case of EP in the past seven years (February 2018 to April 2024).

## Eligibility criteria

Documents from all pregnant mothers who were admitted with a diagnosis of ectopic pregnancy and treated with ectopic pregnancies were included. Documents of patients who were lost from the record office during data collection were not included and incomplete documents of patients who lacked the date of admission and discharge were not included.

## Variables of study

**Dependent variable.** The outcome of ectopic pregnancy

**Independent variable. Sociodemographic characteristics:** Address, age, educational status, ethnicity, marital status, occupation, and religion

**Obstetric/surgical, Gynecologic/contraceptives, and Behavioral factors:** Abortion history, alcohol drinking, cigarette smoking, gravidity, history of appendectomy, history of cesarean section, history of contraceptive use, history of ectopic pregnancy, history of STIs/STD, and history of tubal surgery

## Operational definitions & terminology

1. Ectopic pregnancy: implantation of fertilized ovum at any site outside the endometrial lining of the uterus, in which the patient presents with abnormal vaginal bleeding, abdominal pain, and amenorrhea.

2. Abortion: termination of pregnancy before it reaches viability (<28 weeks)

3. Pelvic inflammatory disease (PID is a spectrum of inflammatory disorders of the upper female genital tract above the internal cervical os.

4. Sexually transmitted disease: infections established through sexual contact.

5. Amenorrhea: the absence of menstruation

6. Appendectomy: surgical removal of the appendix

7. Salpingectomy: surgical removal of fallopian tubes

8. Oophorectomy: surgical removal of the ovary

9. Hysterectomy: surgical removal of the uterus

10. Gravity: number of pregnancies, including the present one, irrespective of the outcomes

11. Shock: a state of deranged vital signs with systolic B/P<90/60 mmHG [1,4,19]

## Sample size and sampling procedure

Records of all pregnancies and gynecological admissions to AURH in the seven years from February 1, 2018, to April 21, 2024), were reviewed. All records of women who had an ectopic pregnancy and who were managed for an ectopic pregnancy from February 1, 2018, to April 21, 2024, who were eligible were included in the study and used for data analysis. A medical record number (MRN) of all the patients admitted to the MCH, gynecological and obstetric ward was collected from the registration book of a hospital. Then, from the collected medical records, the medical record numbers (MRNs) of all patients managed with an ectopic pregnancy were identified. Purposive, nonprobability sampling techniques were applied. Then, the cards were searched in the patient's card room by the data clerk. All the data were collected by trained data collectors.

## Data collection tool, data processing and analysis and data quality assurance

The standardized checklist was used. It was developed and modified in a local context on the basis of a review of the relevant literature. Before the actual data collection, the checklist was pretested on 5% of pregnant women at Wollega University referral hospitals. Then, necessary corrections were made on the basis of the results of the pretest of the instrument. Data were collected via Kobo Toolbox version 2022.4.4, a powerful and reliable software used for data collection and management. Skip logic and validation criteria in the Kobo Toolbox account were used to control the quality of data collection. Training about the Kobo Toolbox was given to the data collectors. The data were collected by one trained midwife and one nurse. One health officer was the supervisor. The questionnaire was checked for completeness and internal consistency. The data were processed via SPSS version 26 software and summarized via statistical procedures, and descriptive statistics were analysed. Multivariate linear logistic regression analysis was executed to select candidate variables for multivariable logistic regression to identify the predictors. Variables with a p-value of less than 0.25 were selected for multivariable logistic regression, and 95% confidence intervals (CIs) were used to describe the associations between the outcome of ectopic pregnancy and potential risk factors. Variables with a p-value < 0.05 in the multivariable analysis were considered significant risk factors for ectopic pregnancy. The quality of the data was ensured during data collection, coding, export, and analysis. During data collection, adequate training and close follow-up were given to the data collectors and supervisors. Incomplete checklists were returned to the data collector for completion.

## Ethical declarations

**Ethics approval and consent to participate.** Ethical approval or an official permission letter was obtained from the Department Research and Ethics Review Committee (DREC) of the Department of Anatomy, Meeting No.DRERC/13/20/2024, the College of Health Science (CHS), and Addis Ababa University (AAU). A permission letter (a formal letter of cooperation) was written to the Ambo University referral hospital (AURH) administration office, and the study was started after receiving formal permission from them. The Declaration of the Institutional Review Boards that authorized the study was followed in the execution of all procedures.

## Disclosure of ethics statement

Formal informed consent form was read and signed to perform all procedures in accordance with a relevant guidelines and regulations to protect human subjects and their data. The confidentiality of the patients was maintained. The names and personal identifiers of the participants were not exposed.

## Results

### Magnitude of ectopic pregnancy

During the period under review from February 2018 to April 2024, 6,249 gynecologic admissions and a total of 17,687 pregnancies were recorded. Of these, 182 were diagnosed with ectopic pregnancy, among which 173 were fit for data analysis. However, the remaining 9 patients were excluded on the basis of the exclusion criteria. The overall magnitude of ectopic pregnancy in this study was 0.98% (173 of 17,687) of pregnancies, which accounted for 2.77% (173 of 6,249) of all gynecologic admissions during the study period.

### Factors associated with ectopic pregnancy

#### Sociodemographic characteristics

As indicated in Table 1. Below, the majority of the cases 121 (69.9%) were urban residents. The majority of the patients 81 (46.8%) were within the age group of 25–29 years, with a mean age of 27.16 (SD ± 4.77) years. The majority of the patients were married 98 (56.6%), and 55 (31.8%) were single. As shown in Fig 2, the majority of the participants included

**Table 1. Sociodemographic characteristics of patients with ectopic pregnancy at Ambo University referral hospital (from February 2018 to April 2024).**

| Sociodemographic characteristics | Frequency (n = 173) | Percent (100%) |
|---|---|---|
| Age group | | |
| ≤19 | 5 | 2.9 |
| 20-24 | 45 | 26.0 |
| 25-29 | 81 | 46.8 |
| 30-34 | 22 | 12.7 |
| 35-39 | 18 | 10.4 |
| 40 and above | 2 | 1.2 |
| Mean = 27.16 ± 4.77SD | | |
| Marital status | | |
| Married | 98 | 56.6 |
| Widowed | 5 | 2.9 |
| Divorced | 15 | 8.7 |
| Single | 55 | 31.8 |
| Educational status | | |
| Unable to read and write | 29 | 16.8 |
| 1-8 Grade | 31 | 17.9 |
| 9-12 Grade | 58 | 33.5 |
| College and Above | 55 | 31.8 |
| Residence | | |
| Rural | 52 | 30.1 |
| Urban | 121 | 69.9 |
| Occupation | | |
| Housewife | 79 | 45.7 |
| Government employee | 14 | 8.1 |
| Merchant | 20 | 11.6 |
| Daily laborer | 5 | 2.9 |
| Others | 55 | 31.8 |
| Religion | | |
| Orthodox | 91 | 52.6 |
| Muslim | 26 | 15.0 |
| Protestant | 52 | 30.1 |
| Other | 4 | 2.3 |
| Ethnicity | | |
| Oromo | 65 | 37.6 |
| Amhara | 54 | 31.2 |
| Tigre | 24 | 13.9 |
| Gurage | 26 | 15.0 |
| Others | 4 | 2.3 |

**SD =** Standard deviation

in this study were Oromo 65 (37.6%) and Amhara 54 (31.2%) (**Fig 2**). With respect to education level, approximately 58 (33.5%) participants had completed high school, and 55 (31.8%) participants had completed college above education or above. Most of the women in this study were housewives 79 (45.7%) and merchants 20 (11.6%) by occupation. Approximately 91 (52.6%) women were Orthodox and 52 (30.1%) women were Protestant by religion (Table 1).

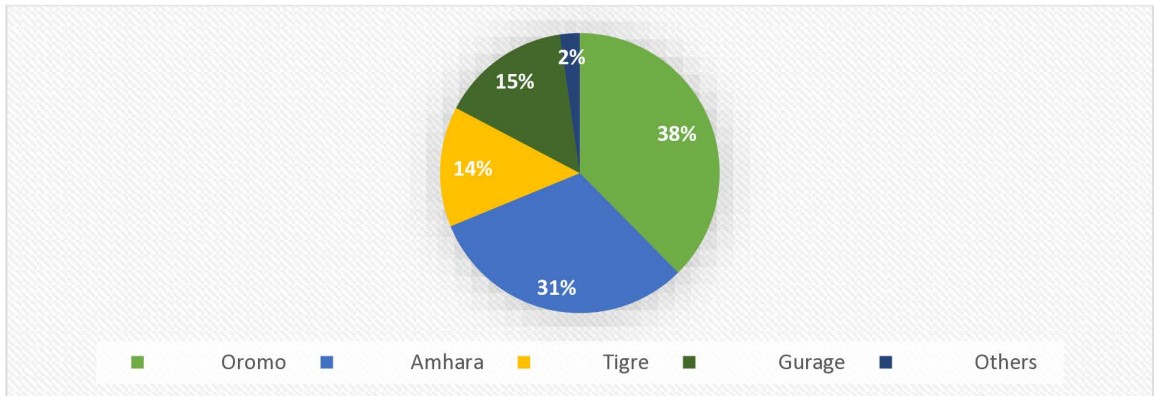

**Fig 2. Ethnicity of patients with ectopic pregnancy at Ambo University referral hospital (from February 2018 to April 2024).**

## Management outcome of ectopic pregnancy

Gestational age (GA) at presentation and parity are shown in Table 2. The majority of the patients, 149 (86.1%) had a history of amenorrhea of < 7 weeks, and 18 (10.4%) patients had a history of amenorrhea of 7_9 weeks. The mean gestation time was 1.2 SD ± 0.46. Among the reviewed patients' cards, the majority of the patients, 51 (29.5%), were nulliparous, and 40 (23.1%) of the participants were multiparous. The mean parity was 2.95 SD ± 1.72 (Table 2**).**

### Mode of presentation

Fig 3 depicts the patients' clinical appearance. The most prevalent chief complaint that patients complained of was abdominal pain, which was reported by 77 (44.5%) patients, followed by abnormal vaginal bleeding in 53 (30.6%) patients. Only 6 (3.5%) of the patients had syncope as their primary complaint, whereas 37 (21.4%) of the patients had amenorrhea (Fig 3**).**

### Diagnosis of ectopic pregnancy

In this study, a urine pregnancy (HCG) test was performed on all the women, and the results were all positive. Among these, 94 cases (54.3%) of ectopic pregnancies were identified via a combination of clinical, culdocentesis, and ultrasound

**Table 2. Gestational age and parity of patients with ectopic pregnancy at Ambo University referral hospital (from February 2018 to April 2024).**

| Gestational age in weeks | Frequency (n = 173) | Percent (100%) |
|---|---|---|
| <7 weeks | 149 | 86.1 |
| 7- 9 weeks | 18 | 10.4 |
| >9 weeks | 6 | 3.5 |
| Parity | | |
| 0 | 51 | 29.5 |
| 1 | 28 | 16.2 |
| 2 | 40 | 23.1 |
| 3 | 9 | 5.2 |
| 4 | 22 | 12.7 |
| >5 | 23 | 13.3 |

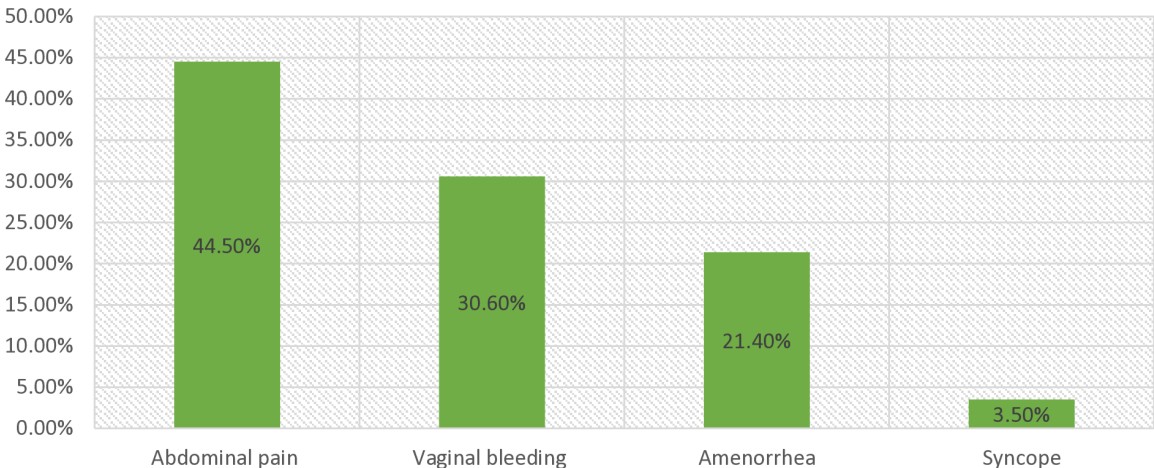

**Fig 3. Clinical presentation of a patient with an ectopic pregnancy at Ambo University referral hospital (from February 2018 to April 2024).**

**Table 3. Diagnosis of patients with ectopic pregnancy at Ambo University referral hospital (from February 2018 to April 2024).**

| Diagnosis | Frequency(n = 173) | Percentage (100%) |
|---|---|---|
| Urine HCG | 173 | 100.0 |
| Clinical and U/S | 65 | 37.6 |
| Clinical and culdocentesis | 5 | 2.9 |
| Clinical, Culdocentesis, and U/ S | 94 | 54.3 |
| Intraoperative | 9 | 5.2 |
| Total | 173 | 100.0 |
| Laterality of EP | | |
| Right-side FT | 144 | 83.2 |
| Left-side FT | 29 | 16.8 |
| Total | 173 | 100.0 |

EP=Ectopic pregnancy, FT=Fallopian tube

scans; another 65 cases (37.6%) were identified via a combination of clinical and ultrasound scans. Intraoperatively, ectopic pregnancies cases were discovered in only 9 (5.2%) patients (Table 3).

In our study, the right-sided fallopian tube accounted for 144 (83.2%) of the ectopic pregnancy cases, whereas the left-sided fallopian tube accounted 29 (16.8%) cases. It is shown both in Fig 4 below and in Table 4 below (Table 4). Table 4 below shows the site of ectopic pregnancy, and it was found that tubal ectopic pregnancy was the most common site accounting for 91.9% (159) of these cases, followed by ovarian ectopic pregnancy accounting for 5.2% (9) of these cases, and only one case of cervical ectopic pregnancy was recorded. In this study, 160 (92.49%) cases were recorded as ruptured ectopic pregnancies. Thirteen (7.5%) cases were recorded as unruptured ectopic pregnancies. It is shown in **Fig 5** below. As shown in Fig 6, there were 106 cases (61.3%) of tubal ectopic pregnancies, of which 33 cases (19.1%) were isthmic; 19 cases (11%) were infundibulum ectopic pregnancies; 12 cases (6.9%) were fimbrial ectopic pregnancies; and the fewest cases (3.7%) were cornual ectopic pregnancies (Fig 6).

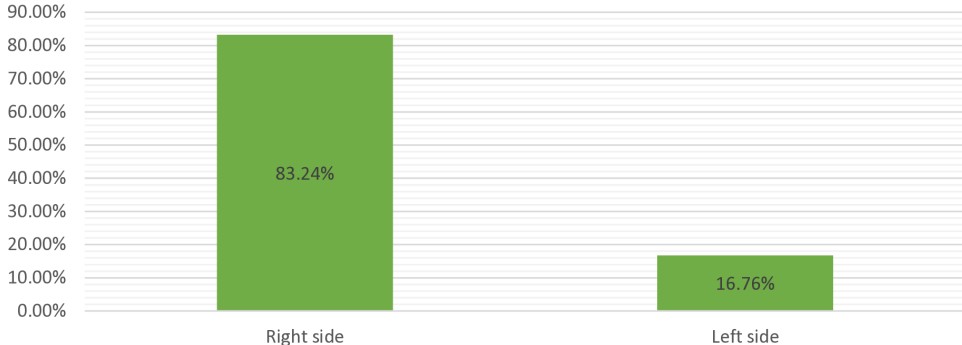

**Fig 4. Laterality of ectopic pregnancy at Ambo University referral hospital (from February 2018 to April 2024..**

**Table 4. Type and site of ectopic pregnancy at Ambo University referral hospital (from February 2018 to April 2024).**

| Type | | Frequency(n=173) | Percent (100%) |
|---|---|---|---|
| | Tubal | 159 | 91.9 |
| | Abdominal | 3 | 1.7 |
| | Ovarian | 9 | 5.2 |
| | Cervix | 1 | .6 |
| Site of EP | | | |
| | Ampulla | 106 | 61.3 |
| | Isthmus | 33 | 19.1 |
| | Fimbrie | 12 | 6.9 |
| | Infundibulum | 19 | 11.0 |
| | Cornual | 3 | 1.7 |
| Total | | 173 | 100.0 |

EP=Ectopic pregnancy

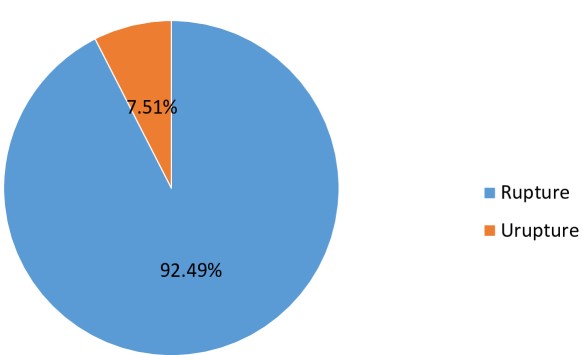

- ■ Rupture
- ■ Urupture

**Fig 5. Condition of ectopic pregnancy at Ambo University referral hospital (from February 2018 to April 2024).**

## Treatment modality of ectopic pregnancy

All 173 women who were diagnosed with ectopic pregnancy underwent surgical treatment, as shown in Fig 7 below. The most common treatment for ectopic pregnancy was salpingectomy, which was used in 133 patients (76.9%). Other treatments included salpingo-oophorectomy in 31 patients (17.9%), oophorectomy in 6 patients 3.5%, and cornual resection in 3 patients (1.7%) (Fig 7).

As indicated in **Table 5** above, the hemoglobin level was > 10 g/dl in 71.1% (123) of patients and 5–10 g/dl in 27.2% (47) of patients. The dominant blood group in this study was A, accounting for 94 (54.3%) of the patients, followed by B and O, accounting for 65 (37.6%) and 14 (8.1%) respectively. The estimated intraoperative blood loss was < 500 ml in 99 (57.2%) patients, and 500 ml–1000 ml in 64 (37.0%) patients. Regarding postoperative complications, only 4 (2.3%) patients developed wound infections, whereas 6 (3.5%) cases developed urinary tract infections (Table 5). Table 6 Below are the gynecologic, obstetric, and behavioral factors of patients with ectopic pregnancy. The majority of the patients, 62 (35.2%), had a previous history of cesarean section, 57 (32.9%) had a history of recurrent STIs or STDs, and 17 (9.8%) had a previous history of pelvic inflammatory disease. However, 53 (30%) of the patients had a history of injectable contraceptive

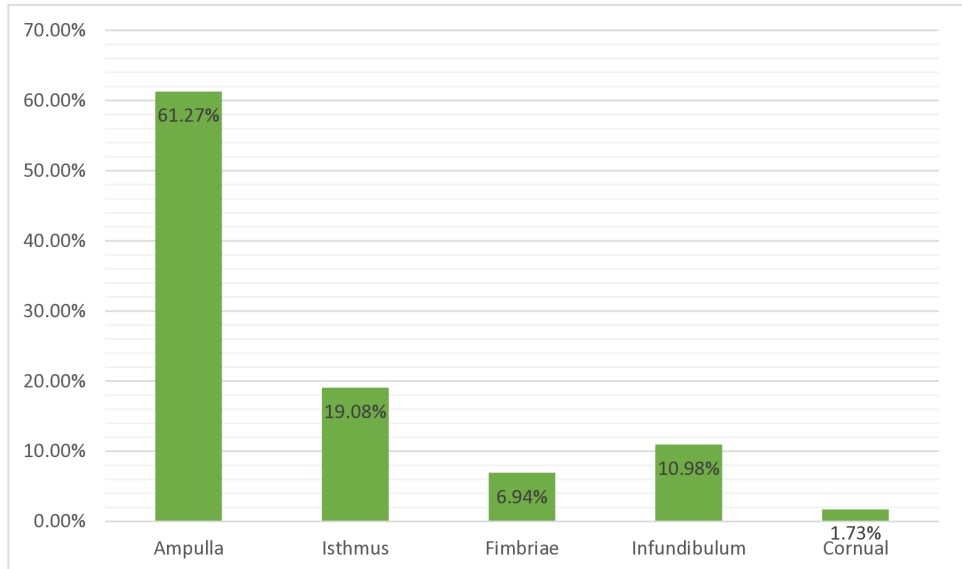

**Fig 6. The site of ectopic pregnancy at Ambo University referral hospital (from February 2018 to April 2024).**

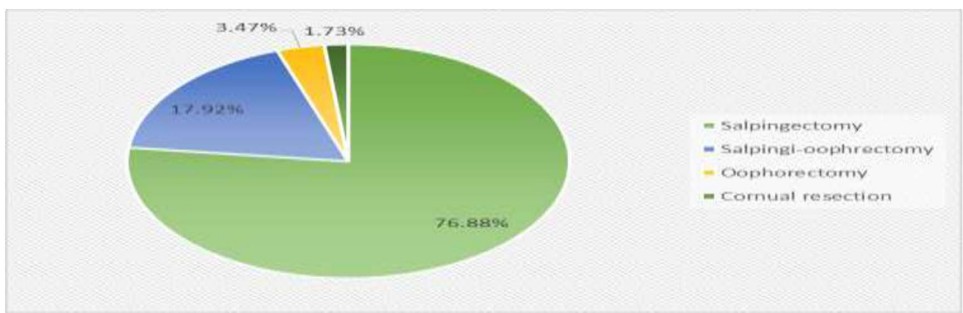

**Fig 7. Treatment modality of ectopic pregnancy.**

**Table 5. Hemoglobin level, blood loss, blood group, and postoperative complications of patients treated with ectopic pregnancy at Ambo University Referral Hospital (from February 2018 to April 2024).**

| Categories | Frequency (N = 173) | Percent (100%) | |
|---|---|---|---|
| Hgb level | <5g/dl | 3 | 1.7 |
| | 5-10g/dl | 47 | 27.2 |
| | >10g/dl | 123 | 71.1 |
| Blood group | | | |
| | A | 94 | 54.3 |
| | B | 65 | 37.6 |
| | O | 14 | 8.1 |
| | | | |
| Blood loss | | | |
| | <500ml | 99 | 57.2 |
| | 500ml -1000ml | 64 | 37.0 |
| | 1000ml - 1500ml | 7 | 4.0 |
| | >1500ml | 3 | 1.7 |
| Hospital stay | | | |
| | < 7 days | 89 | 51.4 |
| | 7-14 days | 75 | 43.4 |
| | > 14 days | 9 | 5.2 |
| Postoperative complications | | | |
| | Anemia | 33 | 19.1 |
| | Postoperative fever | 15 | 8.7 |
| | Wound infection | 4 | 2.3 |
| | Urinary tract infection | 6 | 3.5 |
| | None | 115 | 66.5 |
| Total | | 173 | 100.0 |

methods, and 38 (22.0%) of the patients were ECP users. There were 9 (5.2%), 7 (4.0%), and 9 (5.2%) patients with a previous history of ectopic pregnancy and a history of tubal surgery respectively. Regarding the cigarette smoking habits of the patients, 4 (2.3%) patients were passive cigarette smokers, and 3 (1.7%) patients were active cigarette smokers. Among those patients 1 (0.6%) had a smoking history of less than or equal to 3 years, whereas 2 (1.2%) had a smoking history of greater than or equal to 4 years. Nine (11.0%) of the patients had a history of alcohol consumption (Table 6).

## Logistic regression

### Binary logistic regression analysis

Variables with a p-value < 0.25) were found to be significant in the bivariate analysis. Therefore, mothers who had a previous history of abortion [COR 8.42, 95% CI 1.43–17.21], a history of STIs [COR 3.99, 95% CI 1.53–16.31], a history of PID [COR 6.33, 95% CI 1.36–19.32], a history of ectopic pregnancy [COR 16.22, 95% CI 3.16–24.62], history of tubal surgery [COR 21.91, 95% CI 6.19–36.14], and cigarette smoking [COR 11.68, 95% CI 3.78–19.04] were significantly associated with outcomes of ectopic pregnancy (Table 7).

### Multivariable logistic regression analysis

Variables with a p-value < 0.25 in the bivariate logistic regression analysis were selected as candidate variables for multivariate logistic regression. Therefore, multivariate logistic regression analysis was performed to identify which variables

**Global Public Health**

**Table 6. Gynaecologic, obstetric, and behavioural factors of patients treated with ectopic pregnancy at Ambo University referral hospital (from February 2018 to April 2024).**

| Characteristics | | Frequency | Percent (100%) |
|---|---|---|---|
| Hx of Cesarean section | Yes | 62 | 35.8 |
| | No | 111 | 64.2 |
| Hx of recurrent STIs/STDs | Yes | 57 | 32.9 |
| | No | 116 | 67.1 |
| PID | Yes | 17 | 9.8 |
| | No | 156 | 90.2 |
| Injectable contraceptive users | Yes | 53 | 30.6 |
| | No | 120 | 69.4 |
| ECP users | Yes | 38 | 22.0 |
| | No | 135 | 78.0 |
| Implanon users | Yes | 37 | 21.4 |
| | No | 136 | 78.6 |
| IUCD users | Yes | 20 | 11.6 |
| | No | 153 | 88.4 |
| OCP users | Yes | 13 | 7.5 |
| | No | 160 | 92.5 |
| Hx of ectopic pregnancy | Yes | 9 | 5.2 |
| | No | 164 | 94.8 |
| Hx of tubal surgery | Yes | 7 | 4.0 |
| | No | 166 | 96.0 |
| Hx of appendectomy | Yes | 9 | 5.2 |
| | No | 164 | 94.8 |
| Hx of Infertility | Yes | 9 | 5.2 |
| | No | 164 | 94.8 |
| Hx of abortion | Yes | 14 | 8.1 |
| | No | 159 | 91.9 |
| Hx of cigarette smoking | Nonsmoker | 166 | 96.0 |
| | Passive smokers | 4 | 2.3 |
| | Active smokers | 3 | 1.7 |
| Hx of alcohol consumption | Yes | 19 | 11.0 |
| | No | 154 | 89.0 |
| Total | | 173 | 100.0 |

**Hx** = History

were associated with and predict unfavourable outcomes of ectopic pregnancy. Variables with a p-value < 0.05 in multivariate logistic regression were found to be significant. As a result, variables such as PID, STIs, previous history of ectopic pregnancy, previous history of abortion, previous tubal surgery, and parity were found to have significant associations with EP management outcomes via multivariate logistic regression analysis. Mothers who had a previous history of PID were five times more likely to have unfavourable outcomes from ectopic pregnancy than those with no PID (AOR 5.23, 95% CI 0.28–15.19). Those mothers with a previous history of ectopic pregnancy were eleven times more likely to have ectopic pregnancy than those who had no previous history of ectopic pregnancy (AOR 11.34, 95% CI 6.12–21.74). Those mothers who had a previous history of abortion were six times more likely to have an ectopic pregnancy than those who had no previous history of abortion (AOR 6.29, 95% CI 0.16–12.28). Mothers who had a previous history of tubal surgery were

**Table 7. Bivariate analysis of factors associated with the management outcome of ectopic pregnancy at Ambo University referral hospital (from February 2018 to April 2024).**

| Variable | | EP management outcome | | COR 95% CI | P value |
|---|---|---|---|---|---|
| | | **Favourable** | **Unfavourable** | | |
| Residence | Urban | 103 (85.1%) | 18 (14.9%) | 1 | |
| | Rural | 23 (44.2% | 29 (55.8%) | 7.21(1.76- 26.12) | 0.416 |
| History of STIs | No | 97(83.6%) | 19 (16.4%) | 1 | |
| | Yes | 32 (56.1%) | 25 (43.9%) | 3.99 (1.53- 16.31) | 0.018* |
| History of PID | No | 121 (77.6%) | 35(22.4%) | 1 | |
| | Yes | 6 (35.3%) | 11 (64.7% | 6.33 (1.36- 19.32) | 0.002* |
| History of EP | No | 146 (89%) | 18 (11%) | 1 | |
| | Yes | 3 (33.3%) | 6 (66.7) | 16.22 (3.16 –24.62) | 0.03* |
| History of tubal surgery | No | 149(89.8%) | 17(10.2%) | 1 | |
| | Yes | 2 (28.6%) | 5 (71.4%) | 21.91 (6.19- 36.14) | 0.0032* |
| History of infertility | No | 140 (85.4%) | 24 (14.6%) | 1 | |
| | Yes | 2 (22.8%) | 7 (77.8%) | 20.41 (6.17-48.15) | 0.827 |
| History of abortion | No | 131(82.4%) | 28 (17.6%) | 1 | |
| | Yes | 5 (35.7) | 9 (64.3%) | 8.42(1.43 -17.21) | 0.01* |
| Cigarette smoking | No | 149 (89.8%) | 17 (10.2%) | 1 | |
| | Yes | 3 (42.9%) | 4 (57.1%) | 11.68 (3.78-19.04) | 0.002* |
| Gestational age | < 7 Weeks | 129 (86.6%) | 20 (13.4%) | 1 | |
| | 7-9 Weeks | 11 (61.1%) | 7 (38.9%) | 4.10 (1.37- 9.38) | |
| | >9 Weeks | 2 (33.3%) | 4 (66.7%) | 1.57 (0.63 -6.14) | |
| Hemoglobin level | <5g/dl | 1 (33.3%) | 2(66.7%) | 2.81 (0.71-5.91) | |
| | 5-10g/dl | 12(25.5%) | 35 (74.5%) | 1.34 (0.65- 4.12) | 0.386 |
| | >10g/dl | 112 (91.1%) | 11(8.9%) | 1 | |
| Parity | 0 | 45 (88.2%) | 6 (11.8%) | 1 | |
| | 1 | 19 (67.9%) | 9 (32.1%) | 3.55 (0.71-5.13) | 0.004* |
| | 2 | 19 (37.5%) | 21 (62.5%) | 1 | |
| | 3 | 2 (22.2%) | 7 (77.8%) | 3.17 (0.77-11.78) | |
| | 4 | 10 (45.5%) | 12 (55.5%) | 1 | |
| | >5 | 4 (17.4%) | 19(82.6%) | 3.96 (1.27-9.36) | 0.026* |

**= For variables showing significant association during multivariate analysis (P ≤0.05)

*= For variables showing significant association during bivariate analysis (P ≤ 0.25)

thirteen times more likely to have an ectopic pregnancy than those with no previous history of tubal surgery were (AOR 13.41, 95% CI 4.37–23.82). Mothers who had a history of STIs were three times more likely to have an ectopic pregnancy than those with no STIs were (AOR 3.15, 95% CI 1.28–11.13) (Table 8).

## Discussion

This institutional-based retrospective study aimed to determine the magnitude, management outcomes, and factors associated with the management outcome of ectopic pregnancies treated at Ambo University referral hospital, Oromia region of Ethiopia. The magnitude of ectopic pregnancy in this study was 0.98% (98 out of 10,000 pregnancies); this finding was consistent with studies at Adigrat Hospital in Tigray, Ethiopia (0.82%) [4], with studies at El-Galaa Hospital, Egypt (0.62% and 0.72%) [13], 0.89% at a tertiary hospital in Eastern Nigeria [20], and 0.97% in India [8]. These studies were conducted with the same design, a cross-sectional retrospective study, and a similar study setting, at teaching and referral hospitals

**Table 8. Multivariate analysis of factors associated with the management outcome of ectopic pregnancy at Ambo University referral hospital (from February 2018 to April 2024).**

| Variable | | Management outcome | | AOR 95% CI | P - value |
|---|---|---|---|---|---|
| | | Favourable | Unfavourable | | |
| History STIs | No | 97(83.6%) | 19 (16.4%) | 1 | |
| | Yes | 32 (56.1%) | 25 (43.9%) | 3.15 (1.28- 11.13) | 0.2012** |
| History of PID | No | 121 (77.6%) | 35(22.4%) | 1 | |
| | Yes | 6 (35.3%) | 11 (64.7% | 5.23 (0.28 -15.19) | 0.0021** |
| History of EP | No | 146 (89%) | 18 (11%) | 1 | |
| | Yes | 3 (33.3%) | 6 (66.7) | 11.34(6.12 -21.74) | 0.0312** |
| History of abortion | No | 131(82.4%) | 28 (17.6%) | 1 | |
| | Yes | 5 (35.7) | 9 (64.3%) | 6.29 (0.16- 12.28) | 0.019** |
| Cigarette smoking | No | 149 (89.8%) | 17 (10.2%) | 1 | |
| | Yes | 3 (42.9%) | 4 (57.1%) | 5.17(1.23- 14.23) | 0.2768* |
| History of tubal surgery | No | 149(89.8%) | 17(10.2%) | 1 | |
| | Yes | 2 (28.6%) | 5 (71.4%) | 13.41(4.37-23.82) | 0.0038** |
| Parity | Nulliparous | 45 (88.2%) | 6 (11.8%) | 1 | |
| | ≥1 | 54 (44.3%) | 68 (55.7%) | 9.44 (0.68 -13.16) | 0.001** |

**=** For variables showing significant association during multivariate analysis (P≤0.05)

*= For variables showing significant association during bivariate analysis (P ≤0.25)

where more complicated cases are managed. However, this finding was lower than the 2.2% reported in studies in South Africa [21], 2.05% reported in the Volta Region of Ghana [22], 1.89% reported in the University Hospital of Benin [9], 1.5% reported studies in Adama, Ethiopia [23], 1.3% reported studies in Nnamdi Azikiwe University, Nigeria [24], and 1.1% reported in Nigeria [25]. Furthermore, these results were greater than those of the studies conducted in the Gambia (43 cases; 0.2%), Egypt (2018) (45 cases; 0.51%), and India (100 cases; 0.52%) [13,26,27]. This is because, in this particular study setting, the sample size was relatively large. Moreover, these disparities might be related to differences in environmental factors (exposure to toxic materials), the distribution of related infections (sexually transmitted infections), and behavioral factors (smoking). Therefore, such factors are known to increase the risk of ectopic pregnancy. Smoking can delay the passage of the fertilized ovum into the endometrial lining and its implantation by altering the tubal motility, which can impair the immunity of women, thus increasing their susceptibility to infection. Furthermore, studies indicate that smoking can modify the epithelial cell turnover in the fallopian tube, which increases cellular proliferation and decreases cell death, resulting in structural changes in the epithelial cell surface structure [7,28]. Additionally, exposure to diethylstilbestrol (DES) during gestation increases the chance of ectopic pregnancy because of abnormal tubal morphology and potentially compromised fimbrial function [1,29].

Thus, according to a South African study, up to 86% of women who experienced ectopic pregnancies had evidence of previous genital tract infections; similarly, 42.95% of women in Benin and 35.5% of women in Nigeria had similar evidence. In this study, however, 32.5% of women who experienced ectopic pregnancies had previously experienced genital infections. Furthermore, 0.67% of women in Benin who had ectopic pregnancies had a history of ovulation induction, which was not found in the current study. It can affect the transfer of tubal embryos. However, the exact mechanism is unclear. Research has revealed that infection and smoking hinder the movement of oocytes and embryos via the oviduct by destroying several cilia [30,31].

The majority of the women in this study 98 (56.6%) were married, and 81 (46.8%) were between the ages of 25 and 29 years, with a mean age of 27.16 (SD±4.77) years. This finding was comparable to studies performed in Nepal, Nigeria,

and Benin [9,32]. This could be a consequence of the fact that the women in this decade are experiencing peak periods of fertility and sexual activity. Moreover, young women are biologically vulnerable to sexually transmitted infections because the columnar epithelium extends from the endocervical canal to the ectocervical canal, which makes them fully exposed to pathogens like *Chlamydia trachomatis*, *Neisseria gonorrhoeae*, Mycoplasma, and mixed aerobes and anaerobes [26,29].

Previous ectopic pregnancy, previous tubal surgery, and pelvic inflammatory disease were major risk factors in this study. Hence, mothers with a previous history of ectopic pregnancy were eleven times more likely to have an ectopic pregnancy than those who had no previous history of ectopic pregnancy. Mothers who had a previous history of PID were five times more likely to have an ectopic pregnancy than those with no PID, and mothers who had a previous history of abortion were six times more likely to have an ectopic pregnancy than those who had no previous history of abortion. This was similar to studies performed in Tigray, Ethiopia, Lagos, Nigeria, Yaounde, Cameroon, and Benin, Nigeria [17,33]. Additionally, this finding was corroborated by the globally recognized risk factors for the overall increase in the incidence of ectopic pregnancy. This is because pelvic infection may change tubal function, tubal blockage, tubal adhesion, deciliation (a decrease in the number of cilia and the activity of cilia in the fallopian tube), and fimbrial destruction. According to certain studies, a history of chlamydial infection causes the formation of a protein called PROKR2, which increases the likelihood that a pregnancy will involve implantation in a tubes [1,29]. Previous tubal surgery, previous ectopic pregnancy, and abortion can cause tubal and uterine wall damage, alteration of the normal tubal environment, fimbrial phimosis, tubal occlusion, and hydrosalpinx (a condition where fluid and serosa build in the fallopian tube and cause its swelling). In addition, they can also increase the possibility of infections and their impact [1,26,34]. PID in the current study was seen 9.8%, which is in line with 7.5% result from the Mangalore study [35]. Sexually active, fertile women are nearly solely affected by pelvic inflammatory disease (PID), a polymicrobial infections which increases their risk of ectopic pregnancy (EP) by affecting the normal environment of the fallopian tube, turning the normal tissue into scar that lead to tubal blockage which prohibits eggs from entering the uterus [36,37]. The majority of the women in this study were multiparous, with a mean parity of 2.95 SD ± 1.72. This study was supported by a study performed in Nepal, Nigeria, and Pakistan [1,32,38]. However, this was not supported by a study performed in Hamadan, Iran, in which most of the women in that study were nulliparous and primiparous (20). On the other hand, Williams and Obstetrics, 26th edition, asserts that as the number of deliveries increases, so does the probability of ectopic pregnancy [1]. This is because with advancing age and parity, tubal myoelectric activity decreases. As a result of the peristaltic action of the tube slowing down, the zygote implants before it enters the uterine cavity [34,37].

The findings of this study, which are also supported by a population-based study performed in France, assert that multiparous women are more likely to experience an ectopic pregnancy. They are more likely to be exposed to most risks. However, the physiological effect of advanced age at conception on the likelihood of an ectopic pregnancy is unknown. Age-related changes in tubal function may lead to tubal implantation by postponing ovum transfer [39]. Abdominal pain was the most common clinical profile of the patients in this study. Because of the rupture of the fallopian tube, leakage of blood results in intraabdominal blood collection (hemoperitoneum), followed by abnormal vaginal bleeding, amenorrhea, and syncope. This finding was similar to those of studies performed in India, Nigeria, Ethiopia, and Cameroon [25,30,40].

The majority of the ectopic pregnancies recorded ruptured 160 (92.49%) because the growing embryo became too large for the lumen of the tube to hold, causing the tube to break and burst. In addition, the fallopian tube lacks a submucosal layer. The fertilized ovum immediately burrows the epithelium, causing the zygote to lie close to or inside the muscular layer, which is invaded by trophoblasts that proliferate quickly. This agreed with the studies performed in Bangladesh, Nigeria, Ghana, and Gambia [1,16,39,41]

This indicates a delayed diagnosis because of poor health-seeking behavior, a lack of knowledge about the negative impacts of EP, and poverty.

This study revealed that the predominant site of ectopic pregnancy was a tubal ectopic pregnancy 159 (91.9%), and the ampulla was the most commonly affected segment of the tube 106 (61.3%) because of its anatomical nature, which is the

longest and widest segment of the tube where fertilization normally occurs. It is also followed by an isthmus 33 (19.1%). The majority of them were right-sided tubal ectopic pregnancies 144 (83.24%). This finding was similar to those of studies in Nigeria, India, Bangladesh, and South Africa (2, 6, 21, and 40). This was probably due to latent inflammation of the appendix.

In this study, only one case of cervical ectopic pregnancy was recorded, and no deaths were recorded, unlike a study performed in Nigeria [42]. Salpingectomy was the most common treatment for ectopic pregnancy in this study, accounting for 133 (76.9%) of the cases. This was consistent with research conducted in Adigrat, Ethiopia (70.1%), Yaounde, Cameroon (95.6%), Andhra Pradesh, India (72%), and Ilorin, Nigeria (85.1%) [15,17]. This is because the majority of patients present with ruptured ectopic pregnancy and hemoperitoneum, which manifests as vaginal bleeding and abdominal pain. Under such conditions, emergency surgical intervention remains the mainstay of treatment in developing countries such as Ethiopia. On the other hand, research from Italy, Alabama, and Spain showed that, for patients who are hemodynamically stable and fit other requirements, medical care of an ectopic pregnancy can be a safe way to preserve future fertility. But it was followed by a number of complications, including drug side effects, organ damage, gastrointestinal problems, treatment failure that necessitating surgery, and increase sensitivity to sunlight that causes skin problems [1,43,44]. The majority of the 149 (86.1%) patients presented at an estimated gestational age of < 7 weeks in this study. This finding was in line with many other studies performed in Nigeria, Iraq, Saudi Arabia, and Ethiopia [4,33,42]. The majority of the patients, 123 (71.1%), presented with hemoglobin levels > 10 g/dl, and the estimated intraoperative blood loss was < 500 ml in 99 (57.2%) patients in the present study. This finding was similar to those of studies performed in Ethiopia and Benin, [4,9]. The dominant blood groups in this study were A 94 (54.3%) and B 65 (37.6%). This differs from the result of a study performed in Benin, in which the blood O group was the dominant group [9]. This topic needs further research. Anaemia in 33 (19.1%) patients and postoperative fever in 15 (6.7%) patients were the most common postoperative complications. This result was in line with a study performed in Nigeria [20].

## Strengths and limitations of the study

### Strengths.

- This study was conducted with a relatively large sample size compared with other studies

- As a retrospective study, the extracted data were real-world and unbiased.

- As the data were preexisting data (secondary data), the researcher did not influence the outcome of a variable.

- The study was efficient and cost-effective, with rapid data collection.

### Limitations.

- Single center studies

- Another limitation of this study is being a retrospective study. Because it is very prone to incomplete data and, may not record data in the context of the research topic

- Recording bias

- Sample size constraints, the number of cases is limited by rare exposures or results.

- Misclassification, relying on medical records dilute the observed effect sizes.

## Conclusion

The magnitude of the ectopic pregnancy in this study was 0.98%, which is similar to the global range. The majority of the patients attending Ambo University referral hospital for ectopic pregnancy were between the ages of 25 and 29

years, with a mean age of 27.16±4.77 years, were married, and were from urban areas. In this study, the majority of the 149 (86.1%) patients presented at an estimated gestational age of < 7 weeks, 123 (71.1%) presented with a hemoglobin level >10 g/dl, and the estimated intraoperative blood loss was < 500 ml in 99 (57.2%) patients. The right-sided fallopian tube accounted for 144 (83.2%) ectopic pregnancy cases, whereas the left-sided fallopian tube accounted for 29 (16.8%) cases. A total of 160 (92.49%) ruptured ectopic pregnancies were the most common type, with the tube breaking and rupturing because the growing embryo was too large for the lumen to contain. Additionally, anaemia 33 patients; (19.1%) and postoperative fever 15 patients; (6.7%) were the most common postoperative complications reported in this study. The major risk factors identified in this study were previous abortion, PID, a previous history of ectopic pregnancy, and previous tubal surgery. The reproductive performance of women around the globe, which increases maternal mortality and morbidity, remains a major challenge. Attention must be given as it is a pertinent public health issue in our country, Ethiopia.

## Recommendations

This study was institutional-based, single center retrospective research and a small number of cases were obtained from our study area; thus, it was difficult to draw conclusions for the whole population, which represents real images of ectopic pregnancy. However, on the basis of my findings, the following recommendations were are proposed:

1. **Hospital**

- A high index of clinical suspicion is very important for the early diagnosis of ectopic pregnancy.
- It is important to closely monitor women who have had previous abortions, previous ectopic pregnancies, or previous tubal surgeries, and even if they do not exhibit any symptoms, they should always receive counselling regarding the potential hazards of an ectopic pregnancy.
- Health professionals should be encouraged to provide health information on the signs and symptoms of ectopic pregnancy and about the risk factors for ectopic pregnancy.

2. **Policymakers**

- Promoting antenatal care services for timely detection of ectopic pregnancy, screening, and early intervention of risk factors of ectopic pregnancy (STIs, PID) to decrease the incidence of ectopic pregnancy.

3. **Researcher**

- Further research is needed to explore the relationship between blood group and ectopic pregnancy.
- Further research is needed to assess why ectopic pregnancy is most common in the right fallopian tube.
- A large-scale population-based study using primary data to assess all possible risk factors for ectopic pregnancy should be carried out.
- The genetic cause of ectopic pregnancy in embryos should be studied.
  - In general, improving safe motherhood in women's lives requires inclusive commitments at all levels, at the individual level, at home, in the community, and in the country.
  - To minimize its complications, attention must be given, as it is a pertinent public health issue in our country, Ethiopia.

## Supporting information

**S1 Checklist.  English version checklist.**
(DOCX)

**S1 Data.  Dataset.**
(RAR)

## Acknowledgments

We highly extend a debt of gratitude to all participants in this research, supervisors of data collection, and data collectors for their worthy dedication and efforts. Additionally, we are grateful to the administrative bodies at all levels that supported our research and who encouraged us to conduct this study.

## Author contributions

**Conceptualization:** Nigussie Tesfaye Gizaw, Mekbeb Afework K/Mariam, Motuma Gutu Fayera.

**Formal analysis:** Nigussie Tesfaye Gizaw, Mekbeb Afework K/Mariam, Motuma Gutu Fayera.

**Methodology:** Nigussie Tesfaye Gizaw.

**Validation:** Motuma Gutu Fayera.

**Visualization:** Nigussie Tesfaye Gizaw.

**Writing – original draft:** Nigussie Tesfaye Gizaw.

**Writing – review & editing:** Nigussie Tesfaye Gizaw, Mekbeb Afework K/Mariam.

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
