## [Decision Letter · Decision Letter 0]

9 Feb 2025

PGPH-D-24-03013

Magnitude of ectopic pregnancy, management methods, and its associated factors among pregnant women attending Ambo University Referral Hospital in Oromia Regional State, Ethiopia: A seven years retrospective institional based cross-sectional study

Dear Dr. Gizaw,

Thank you for submitting your manuscript to PLOS Global Public Health. After careful consideration, we feel that it has merit but does not fully meet PLOS Global Public Health’s publication criteria as it currently stands. Therefore, we invite you to submit a revised version of the manuscript that addresses the points raised during the review process.

The manuscript has been evaluated by two reviewers, and their comments are available below. Could you please carefully revise the manuscript to address all comments raised?

We look forward to receiving your revised manuscript.

Kind regards,

Jianhong Zhou

Staff Editor

Comments from PLOS Editorial Office: We note that one or more reviewers has recommended that you cite specific previously published works. As always, we recommend that you please review and evaluate the requested works to determine whether they are relevant and should be cited. It is not a requirement to cite these works. We appreciate your attention to this request.

Additional Editor Comments (if provided):

Reviewers' comments:

Reviewer's Responses to Questions

**Comments to the Author**

1. Does this manuscript meet PLOS Global Public Health’s publication criteria ? Is the manuscript technically sound, and do the data support the conclusions? The manuscript must describe methodologically and ethically rigorous research with conclusions that are appropriately drawn based on the data presented.

Reviewer #1: Partly

Reviewer #2: Yes

2. Has the statistical analysis been performed appropriately and rigorously?

Reviewer #1: Yes

Reviewer #2: Yes

3. Have the authors made all data underlying the findings in their manuscript fully available (please refer to the Data Availability Statement at the start of the manuscript PDF file)?

Reviewer #1: Yes

Reviewer #2: Yes

4. Is the manuscript presented in an intelligible fashion and written in standard English?

Reviewer #1: Yes

Reviewer #2: Yes

5. Review Comments to the Author

Reviewer #1: The authors present a manuscript which aims to assess the seven-year-long experience of a single health institution about ectopic pregnancy. The study has been conducted properly and the manuscript has been well written but several corrections should be made to achieve better comprehension. First, the introduction part is too long and it should be condensed into one page. Second, the discussion part is a mere reiteration of what is already known about the risk factors and diagnosis of ectopic pregnancy. Instead, the authors should discuss the clinical implications of their findings and the specific aspects of their findings which distinguish them from previously published similar studies. Third, the authors should mention "retrospective" study design as a power-limiting factor. Lastly, all references that were published before 2008 should be replaced with newer and more up-to-date ones if possible.

I would recommend that the manuscript should be reviewed once more after required corrections have been made.

Reviewer #2: Dear Authors thank you to allow me to review this interesting manuscript. It is a big retrospective study about the treatment and and managemente of different types of ectopic pregnancies. The manuscript is well written and complete and describes all the fundamental aspects about this type of pathology. However, in my opinion the discussion should be expanded talking about the role of PID in development of ectopic pregnancy underlining the pejorative effect of chlamydia compared to other pathogens (PMID: 23007248). Furthermore all the ectopic pregnancies are trteated with a surgical approach. The authors must justify it, writing in the discussion section abuout the possibility of the medical treatment and also of its possible complications (10.3390/diagnostics10090652 ; 10.3390/jcm12237396)

6. PLOS authors have the option to publish the peer review history of their article (what does this mean? ). If published, this will include your full peer review and any attached files.

**Do you want your identity to be public for this peer review?** For information about this choice, including consent withdrawal, please see our Privacy Policy .

Reviewer #1: **Yes: ** Mine Kanat-Pektas, M.D., Professor

Reviewer #2: No

---

## [Decision Letter · Decision Letter 1]

17 Mar 2025

PGPH-D-24-03013R1

Magnitude of ectopic pregnancy, management methods, and its associated factors among pregnant women attending Ambo University Referral Hospital in Oromia Regional State, Ethiopia: A seven years retrospective institional based cross-sectional study

Dear Dr. Gizaw,

Thank you for submitting your manuscript to PLOS Global Public Health. After careful consideration, we feel that it has merit but does not fully meet PLOS Global Public Health’s publication criteria as it currently stands. Therefore, we invite you to submit a revised version of the manuscript that addresses the points raised during the review process.

We look forward to receiving your revised manuscript.

Kind regards,

Miquel Vall-llosera Camps

Staff Editor

Journal Requirements:

Additional Editor Comments:

Please fix your title, it should be:

"Magnitude of ectopic pregnancy, management methods, and its associated factors among pregnant women attending Ambo University Referral Hospital in Oromia Regional State, Ethiopia: A seven years retrospective institutional based cross-sectional study"

Reviewers' comments:

Reviewer's Responses to Questions

**Comments to the Author**

1. If the authors have adequately addressed your comments raised in a previous round of review and you feel that this manuscript is now acceptable for publication, you may indicate that here to bypass the “Comments to the Author” section, enter your conflict of interest statement in the “Confidential to Editor” section, and submit your "Accept" recommendation.

Reviewer #1: All comments have been addressed

Reviewer #2: (No Response)

2. Does this manuscript meet PLOS Global Public Health’s publication criteria ? Is the manuscript technically sound, and do the data support the conclusions? The manuscript must describe methodologically and ethically rigorous research with conclusions that are appropriately drawn based on the data presented.

Reviewer #1: Yes

Reviewer #2: Yes

3. Has the statistical analysis been performed appropriately and rigorously?

Reviewer #1: Yes

Reviewer #2: Yes

4. Have the authors made all data underlying the findings in their manuscript fully available (please refer to the Data Availability Statement at the start of the manuscript PDF file)?

Reviewer #1: Yes

Reviewer #2: Yes

5. Is the manuscript presented in an intelligible fashion and written in standard English?

Reviewer #1: Yes

Reviewer #2: Yes

6. Review Comments to the Author

Reviewer #1: The authors present a very detailed and thorough evaluation of ectopic pregnancies that were diagnosed and treated at a referral hospital during a period of 7 years. Although the study was conducted properly and the manuscript is well written, several corrections should be made to achieve better comprehension. First, the introduction part should be condensed into one page. Second, the authors should mention about the factors that limit the power of their study (i.e. retrospective design, recall/recording bias). Third, the authors should replace the references that were published before 2010 with newer and more up-to-date ones if possible.

Reviewer #2: Dear Authors, Thank you to allow mes to review this interesting manuscript. It is a retrospective analysis on the ectopic pregnancies treated in Ambo University Referral Hospital in Oromia Regional State, Ethiopia. The authors have performed a deep analysis on the risk factors (social and demographics) the type of symptoms, diagnosis and therapies. Also the statistical analysis was correctly performed. The figures are clear. The sample is very large. However i have some questions:

- Why some patients underwent oophorectomy? They had an ovarian ectopic pregnancy? This should be specified

To be more readable I suggest to shorten the discussion section. Furthermore, the manuscript report only surgical treatment and this is justified by the economical and social context. However author correctly report information about the medical treatment. They report complications related to the drug side effects but they aren't the only type of complications. Sometimes are reported also organ damage or other types of complications related to the ectopic pregnancy itself even when this has resolved with medical therapy. I suggest to read and cite 10.3390/jcm12237396.

7. PLOS authors have the option to publish the peer review history of their article (what does this mean? ). If published, this will include your full peer review and any attached files.

**Do you want your identity to be public for this peer review?** For information about this choice, including consent withdrawal, please see our Privacy Policy .

Reviewer #1: **Yes: ** Mine Kanat-Pektas

Reviewer #2: No

---

## [Decision Letter · Decision Letter 2]

21 Apr 2025

Magnitude of ectopic pregnancy, management methods, and its associated factors among pregnant women attending Ambo University Referral Hospital in Oromia Regional State, Ethiopia: A seven years retrospective institutional based cross-sectional study

PGPH-D-24-03013R2

Dear Mr Gizaw,

We are pleased to inform you that your manuscript 'Magnitude of ectopic pregnancy, management methods, and its associated factors among pregnant women attending Ambo University Referral Hospital in Oromia Regional State, Ethiopia: A seven years retrospective institutional based cross-sectional study' has been provisionally accepted for publication in PLOS Global Public Health.

Best regards,

Julia Robinson

Executive Editor

Reviewer Comments (if any, and for reference):

Reviewer's Responses to Questions

**Comments to the Author**

1. If the authors have adequately addressed your comments raised in a previous round of review and you feel that this manuscript is now acceptable for publication, you may indicate that here to bypass the “Comments to the Author” section, enter your conflict of interest statement in the “Confidential to Editor” section, and submit your "Accept" recommendation.

Reviewer #1: All comments have been addressed

Reviewer #2: All comments have been addressed

2. Does this manuscript meet PLOS Global Public Health’s publication criteria ? Is the manuscript technically sound, and do the data support the conclusions? The manuscript must describe methodologically and ethically rigorous research with conclusions that are appropriately drawn based on the data presented.

Reviewer #1: Yes

Reviewer #2: Yes

3. Has the statistical analysis been performed appropriately and rigorously?

Reviewer #1: Yes

Reviewer #2: Yes

4. Have the authors made all data underlying the findings in their manuscript fully available (please refer to the Data Availability Statement at the start of the manuscript PDF file)?

Reviewer #1: Yes

Reviewer #2: Yes

5. Is the manuscript presented in an intelligible fashion and written in standard English?

Reviewer #1: Yes

Reviewer #2: Yes

6. Review Comments to the Author

Reviewer #1: I would recommend that the revised version of this manuscript can be accepted for publication in PLOS Global Public Health.

Reviewer #2: (No Response)

7. PLOS authors have the option to publish the peer review history of their article (what does this mean? ). If published, this will include your full peer review and any attached files.

**Do you want your identity to be public for this peer review?** For information about this choice, including consent withdrawal, please see our Privacy Policy .

Reviewer #1: **Yes: ** Mine Kanat-Pektas

Reviewer #2: No
